# New Perspectives of Gene Therapy on Polyglutamine Spinocerebellar Ataxias: From Molecular Targets to Novel Nanovectors

**DOI:** 10.3390/pharmaceutics13071018

**Published:** 2021-07-03

**Authors:** Fabiola V. Borbolla-Jiménez, María Luisa Del Prado-Audelo, Bulmaro Cisneros, Isaac H. Caballero-Florán, Gerardo Leyva-Gómez, Jonathan J. Magaña

**Affiliations:** 1Laboratorio de Medicina Genómica, Departamento de Genética, Instituto Nacional de Rehabilitación-Luis Guillermo Ibarra Ibarra, Ciudad de México 14389, Mexico; fvbj@hotmail.com; 2Programa de Ciencias Biomédicas, Facultad de Medicina, Universidad Nacional Autónoma de México, Ciudad de México 04510, Mexico; 3Departamento de Bioingeniería, Escuela de Ingeniería y Ciencias, Tecnológico de Monterrey Campus Ciudad de México, Ciudad de México 14380, Mexico; luisa.delpradoa@gmail.com; 4Departamento de Genética y Biología Molecular, Centro de Investigación y de Estudios Avanzados del Instituto Politécnico Nacional (CINVESTAV), Ciudad de México 07360, Mexico; bcisnero@cinvestav.mx; 5Departamento de Farmacia, Facultad de Química, Universidad Nacional Autónoma de México, Ciudad de México 04510, Mexico; hiram.qfohead@gmail.com; 6Departamento de Farmacia, Centro de Investigación y de Estudios Avanzados del Instituto Politécnico Nacional (CINVESTAV), Ciudad de México 07360, Mexico

**Keywords:** spinocerebellar ataxias, polyglutamine, gene therapy, expanded triplet repeat, antisense and interferent technology, DNA editing systems, nanoparticulate systems, blood–brain barrier

## Abstract

Seven of the most frequent spinocerebellar ataxias (SCAs) are caused by a pathological expansion of a cytosine, adenine and guanine (CAG) trinucleotide repeat located in exonic regions of unrelated genes, which in turn leads to the synthesis of polyglutamine (polyQ) proteins. PolyQ proteins are prone to aggregate and form intracellular inclusions, which alter diverse cellular pathways, including transcriptional regulation, protein clearance, calcium homeostasis and apoptosis, ultimately leading to neurodegeneration. At present, treatment for SCAs is limited to symptomatic intervention, and there is no therapeutic approach to prevent or reverse disease progression. This review provides a compilation of the experimental advances obtained in cell-based and animal models toward the development of gene therapy strategies against polyQ SCAs, providing a discussion of their potential application in clinical trials. In the second part, we describe the promising potential of nanotechnology developments to treat polyQ SCA diseases. We describe, in detail, how the design of nanoparticle (NP) systems with different physicochemical and functionalization characteristics has been approached, in order to determine their ability to evade the immune system response and to enhance brain delivery of molecular tools. In the final part of this review, the imminent application of NP-based strategies in clinical trials for the treatment of polyQ SCA diseases is discussed.

## 1. Introduction

Spinocerebellar ataxias (SCAs) form a clinically and genetically heterogeneous group of autosomal dominant neurodegenerative disorders that display distinctive neuropathological features, including progressive ataxia, loss of overall movement coordination, cerebellar dysarthria, dysmetria, adiadochokinesia and postural tremor [1,2,3]. Commonly, SCAs also present extracerebellar manifestations such as pyramidal and extrapyramidal signs, cognitive dysfunction, ophthalmoplegia, peripheral neuropathy, sleep disorder and dysautonomic alterations [1]. Although the majority of symptoms and signs are shared between SCAs, it is possible to clinically classify them into three different subtypes: autosomal dominant cerebellar ataxia (ADCA) I, ADCAII and ADCA III [4]. Besides ataxia, features of ADCA I include ophthalmoplegia, optic atrophy and extrapyramidal signs; ADCA II is characterized by macular degeneration, in addition to ataxia and extrapyramidal signs, while ADCA III corresponds to cases of “pure” ataxia, with the onset frequently occurring after the fifth decade of life [5,6]. Nevertheless, based on the identification of the causative genes, more than 40 varieties of SCAs have been described thus far, according to the Online Mendelian Inheritance in Man (OMIM) database, distributed by the National Center for Biotechnology (NCBI) [7,8].

The worldwide prevalence of SCAs is 2/100,000 to 43/100,000 [9], with SCA3 being the most common SCA (21%), followed by SCA2 and SCA6 (15% each), SCA1 (6%) and SCA7 (5%) [10]. Interestingly, all these SCAs are caused by an abnormal expansion of CAG triplet repeats located in their respective loci [4,10]. In each SCA, the expanded CAG repeats give rise to a mutant protein bearing an expanded polyglutamine (polyQ) tract [10], whose expression ultimately causes neuronal damage and extensive neurodegeneration [11]. The polyglutamine SCA diseases (polyQ SCAs) include SCA 1, 2, 3, 6, 7 and 17 and dentatorubral–pallidoluysian atrophy (DRPLA). In healthy subjects, the number of CAG repeats is highly polymorphic, and there is a specific number of CAG repeats for each SCA representing the threshold between the normal and the pathological state (Table 1). However, most SCAs have been identified as having pre-mutated alleles, which cause no clinical manifestations but are known to be unstable and expand to a full mutation in subsequent generations [12]. Hence, CAG repeat length determination has permitted accurate DNA-based diagnosis of polyQ SCAs [13]. The phenomenon of anticipation is present in polyQ SCAs, which is characterized by the reduction in the age of disease onset and worsening of symptoms in affected individuals in successive generations. Interestingly, paternal transmission is more likely to be associated with longer repeat expansions than maternal transmission [2,4,14]. Table 1 depicts the main clinical and neuropathological features of polyQ SCAs.

## 2. Molecular Basis of PolyQ SCAs

A growing body of evidence that emerged in recent years suggests that misfolding of polyQ-containing proteins is a unifying molecular mechanism for all polyQ SCAs, which results in the formation of intracellular aggregates/inclusions in specific brain regions and, ultimately, in cell death and disease development [11,23,24,25] (Figure 1A). It is thought that polyQ SCA pathogenesis is primarily mediated by a deleterious gain of function of the polyQ-containing mutant proteins [26]. Thus, toxic downstream effects mediated by polyQ proteins and their corresponding RNA transcripts alter a variety of cellular processes and pathways, including proteostasis, as supported by experiments in cellular and transgenic mouse models of polyQ SCAs [27,28,29].

A key mechanism impaired in virtually all polyQ SCAs is transcriptional regulation. Ataxins can interact with transcription factors to regulate gene expression [30]. Different studies have demonstrated that ataxins (ATXNs) 1, 2, 3 and 7, TATA-binding protein (TBP) and Atrophin-1 (ATN1) are directly involved in transcription, by acting as components of transcriptional regulatory complexes [11]. Ataxins can bind to DNA elements of some transcription factors (TBP and α1ACT), in order to thereby control transcription repression or activation. Alternatively, binding of ataxins to chromatin can modify chromatin complexes engaged in promoter regions (ATXN7/Spt-Ada-Gcn5 acetyltransferase (SAGA) and ATXN3/ Histone deacetylase 3 (HDAC3)/Silencing Mediator of Retinoic Acid and Thyroid Hormone Receptor (SMRT)/nuclear receptor co-repressor (NCOR)); facilitate the assembly of preinitiation transcription complexes (PIC; TBP/TFIID) to deubiquitinate substrates (ATXN3 and ATXN7/SAGA); or participate in RNA metabolism (ATXN1 and ATXN2) [31,32] (Figure 1B). In addition, transcriptome deregulation, due to changes in the expression profile of microRNAs (miRNAs) and non-coding RNAs, has been correlated with specific pathological stages of polyQ SCAs [33,34,35,36], as shown by microarrays and RNA-sequencing analyses. On the other hand, the contribution of nuclear RNA foci, which are composed of mutant RNA transcripts and RNA-binding proteins [37,38], to cell dysfunction and disease phenotype has been recently proposed [39] (Figure 1B). Specifically, aberrant interaction of ataxins or their respective RNA transcripts with alternative splicing factors implies that RNA splicing might be a new previously unrecognized alteration in polyQ SCAs.

Overall, abnormal interaction of ataxins with native partners and their assembly with new partners impair multiple cellular functions, including autophagy, ubiquitin–proteasome degradation, calcium homeostasis, mitochondrial energy production, activation of pro-apoptotic routes and synaptic neurotransmission (Figure 1C) [19,40,41,42,43].

## 3. Pharmacological Therapy Treatment for PolyQ SCAs

It is well known that physical exercise constitutes one of the pillars in treating polyQ SCA patients with neurodegenerative affectations. Such rehabilitation programs significantly improved the cerebellar symptoms of SCA2, SCA3 and SCA7 patients, through motor learning and neural plasticity mechanisms [44,45,46], which consequently has a beneficial effect on their general health condition. Thus, therapeutic exercising, combined with the administration of neuroprotective drugs in prodromal stages, would improve the quality of life of patients.

On the other hand, current pharmacological treatments for polyQ SCAs are limited to supportive care, partially alleviating some clinical manifestations but failing to halt disease progression [47]. Clinical evidence has shown that treatment with levodopa alleviates rigidity/bradykinesia in SCA2 patients with parkinsonism [48], while painful muscle contractions can be ameliorated with magnesium, quinine, mexiletine or high doses of vitamin B [17,41]. The potassium channel modulators chlorzoxazone and riluzole improved cerebellar electrophysiology in SCA2 patients, probably by modulating the excitability and dendritic plasticity of Purkinje cells [49,50,51]. Finally, 4-aminopyridine (4-AP) has been shown to ameliorate the motor coordination deficiency of SCA1 and SCA6 mice [52,53]. Some obstacles in the search for therapeutic compounds include the following: (1) small samples of patients recruited for clinical trials; (2) patients enrolled in clinical trials exhibited high variability of clinical stages; and (3) lack of quantitative clinical variables to evaluate treatment effectiveness. In fact, no drug-based treatment has been approved by the FDA or EMA.

Some alternative approaches have been undertaken to treat motor symptoms; for instance, thalamic and subthalamic stimulation using (^123^I)β-CIT single-photon emission has been successfully tested to alleviate postural tremor (Table 2) [54,55].

Recent advances in elucidating the molecular basis that underlies polyQ SCAs have boosted the design of therapeutic strategies against these disorders [61]. The goal of some of these strategies is to impede the aggregation of polyQ proteins by the following approaches: (1) use of β-sheet structure-destabilizing compounds; (2) identification/generation of small molecules that act as binding competitors to block the assembly between polyQ protein monomers [62,63,64,65]; (3) overexpression of endogenous chaperones to block polyQ protein misfolding and aggregation [66,67,68] (Figure 2A); (4) polyQ protein clearance by enhancing its degradation via the ubiquitin–proteasome and autophagy pathways (Figure 2B). As a proof of concept, it has been shown that activation of autophagy by treatment with either rapamycin (an mTOR pathway inhibitor), temsirolimus, trehalose (autophagy independent of the mTOR pathway) or lithium (reduces Ca^2+^ exit from receptor InsP3R1) exerts a therapeutic effect in cellular and animal models of polyQ SCAs [69,70,71,72]. Likewise, autophagy induction via the administration of 17-(dimethylaminoethylamino)-17-demethoxygeldanamycin (17-DMAG), cordycepin or resveratrol resulted in decreased polyQ protein aggregates and, consequently, in a reduction in oxidative stress [66,73,74]. Furthermore, administration of the drugs benzamil, Y-27632 and H1152 [75,76,77], including the nutraceutics catalpol, puerarin, T- 11 and daidzein, drove the clearance of ataxin-3 and ataxin-7 in cellular models through enhancement of the ubiquitin–proteasome mechanism [78,79]. Owing to the role of the ubiquitin–proteasome system in the clearance of polyQ protein aggregates, several therapeutic molecules that target components of the proteasomal system have been generated. For instance, sulforaphane can up-regulate the expression of all proteasome subunit genes through activation of the transcription factors Nrf1 and Nrf2 [80,81]. In addition, all of the following inhibitors: inhibitor of ubiquitin-specific protease 14 (IU1), an inhibitor of Protein kinase A (S776), GSK690693, S100B inhibitor (TRTK12) and a synthetic molecule (JMF1907), suppressed the formation of polyQ protein intranuclear inclusions in SCA models [79,82,83,84].

Despite the promissory future of the therapeutic drugs in the clinic of polyQ SCA diseases, some limitations need to be overcome before implementing clinical trials. For instance, virtually all these compounds exhibit poor physicochemical properties (solubility, permeability and metabolic stability), which in turn affect their ability to penetrate the blood–brain barrier (BBB) and exhibit central nervous system (CNS) activity [85].

## 4. Gene Therapy Treatment for PolyQ SCAs

It is worth to mention that elimination of polyQ protein aggregates does not warrant a disease cure because of the existence of alternative polyQ SCA pathological mechanisms, including the aberrant function of both polyQ protein monomers and mutant CAG-containing transcripts. Thus, gene therapy strategies aimed to correct the defective gene or neutralize the RNA toxicity offer an interesting alternative to fight polyQ SCAs. In this section, we present an updated review of the molecular strategies toward SCA therapy (see Table 3).

### 4.1. mRNA-Based Technology—Antisense Oligonucleotides

A non-viral strategy to knock-down gene expression is the antisense oligonucleotide (ASO) tool, short, single-strand chemically modified oligonucleotides that selectively bind to complementary mRNA via Watson–Crick base pairs. ASOs can act through different mechanisms, depending on their chemical modifications and design; ASOs can induce RNase H-mediated cleavage of the targeted mRNA [93] or block translation of the corresponding protein (Figure 3A). Intracerebroventricular (ICV) injection of an ASO, named ASO7, in two mouse models of SCA2 resulted in decreased levels of cerebellar *ATXN2* mRNA and protein and improved motor function [94]. Interestingly, treated SCA2 mice recovered the normal firing frequency of Purkinje cells even when treatment was initiated after the motor phenotype onset. A mouse model expressing the human *ATXN3* gene was subjected to ICV injection using 2′-MOE-modified ASOs with a chimeric (gapmer) design. These ASOs could reduce the mutant protein levels by >50% in the diencephalon, cerebellum and cervical spinal cord [95]. The same laboratory carried out a longitudinal study to assess SCA3 mouse motor function. Transgenic mice were treated with ASOs (ASO-5) at 8 weeks of age and longitudinally evaluated for up to 29 weeks. The treatment with ASO-5 reduced the levels of mutant ataxin-3 in a dose-dependent manner, with the greatest decrease observed at 16 weeks of age [96]. Remarkably, the impaired motor function was mitigated via normalization of Purkinje neuron firing frequency and after hyperpolarization. A modified 2′-O-methyl phosphorothioate (CUG)_7_ triplet repeat ASO was able to decrease the mRNA levels of mutant *ATXN1* and *ATXN3* in patient-derived SCA1 and SCA3 fibroblasts, respectively [97]. Furthermore, ICV injection in an SCA1 knock-in mouse using gapmer ASOs resulted in efficient downregulation of *ATXN1* mRNA and protein levels, with the consequent improvement in motor function and survival [98]. Remarkably, mitigation of retinal degeneration, a major distinctive SCA7 symptom, was observed upon a single intravitreal injection of a therapeutic ASO in an SCA7 knock-in mouse, with effective silencing of both wild-type and mutant *ATXN7* alleles, and the consequent decrease in mutant protein aggregates [99].

### 4.2. mRNA-Based Technology—Allele-Specific Antisense Oligonucleotides

The above-mentioned ASO-based experiments failed to discriminate between the wild-type and the mutant alleles. Thus, some efforts have been conducted to specifically knock-down the mutant allele and thereby maintain the vital function of wild-type (WT) ataxin proteins (Figure 3B). The approach requires the CAG tract expansion to be associated with a single-nucleotide polymorphism (SNP) to target and lower mutant allele levels using ASOs. This strategy has been successfully implemented in Huntington’s disease (HD) [87,100]; thus, the avenue to apply it to polyQ SCAs is open [99].

### 4.3. Exon Skipping by ASOs

An alternative ASO-based strategy to remove the expanded CAG tract is exon skipping (Figure 3C). ASOs can induce exon skipping by sterically blocking the binding of splicing factors to pre-mRNAs, maintaining the RNA reading frame and rendering a truncated but functional protein [88,101]. Skipping of exons 9 and 10 of the *ATXN3* pre-mRNA was obtained in fibroblast cultures derived from an SCA3 patient, using 2′-O-methyl-modified ASOs (ASO9.1 and ASO10) that contained a phosphorothioate backbone; upon treatment, a truncated ataxin-3 lacking 72 amino acids (aa) was synthetized [102]. Notably, these authors showed skipping of exon 9 and exon 10 in the cerebellum of a transgenic SCA3 mouse model after seven days of applying an ICV injection with a mixture of two ASOs directed to the *ATXN3* pre-mRNA of the mouse. Likewise, targeting of *ATXN3* exon 10 with the 2′-O-methoxyethylribose ASO resulted in the removal of the expanded CAG tract in both SCA3 mouse model and SCA3 patient-derived fibroblasts, with the subsequent synthesis of a truncated ataxin-3 protein lacking the toxic poly-glutamine region [102]. The truncated protein, whose ubiquitin binding and cleavage activity remained intact, was detectable in the mouse cortex and cerebellum for up to 2.5 months of age. Upon ASO treatment, a decrease in the accumulation of ataxin-3 protein aggregates was found [88], which implies a significant physiological effect of the treatment. Concerning SCA1, skipping of exon 8 was induced by CAG-targeting ASOs in both patient-derived fibroblast cultures and SCA1^154Q/2Q^ transgenic mice [98]. Reduced mutant *ATXN1* protein levels were found in different mouse brain regions after weekly ICV injections, which suggests that this therapy positively impacted mouse physiology [98].

As mentioned before, the ASO-based exon skipping approach has some limitations. In addition to its inability to distinguish between WT and mutant transcripts, off-target effects that might interfere with the alternative splicing of other genes cannot be ruled out. It is expected that the next generation of splicing-skipping ASOs designed against polyQ SCA diseases would specifically hybridize to the expanded allele while keeping the normal allele expression intact.

### 4.4. Non-Allele Interferent Gene Silencing

RNA interference (RNAi) is a cellular mechanism that induces post-transcriptional gene silencing, implicating small RNAs (21–23 nucleotides long) that can regulate gene expression in eukaryotic organisms by promoting the cleavage of target RNAs [103]. Recent progress in elucidating RNAi mechanisms has favored adapting this process to therapeutic applications (Figure 3A) [104].

Treatment of an SCA1 mouse model at 7 weeks old, through injection into midline cerebellar lobules with two different short hairpin RNAs (shRNA), resulted in decreased ataxin-1 aggregates and motor improvement [105]. The same laboratory developed other RNAi-based strategies against SCA1, using adeno-associated vectors (AAV) [104,106]. Finally, these authors extended the RNAi-mediated silencing of *ATXN1* mRNA to adult rhesus monkeys, through injection into the deep cerebellar nuclei, with AAV that expressed an RNAi (miS1) and co-expressed enhanced green fluorescent protein [107]. Upon therapeutic intervention, reduced levels of endogenous *ATXN1* mRNA (~30%) were found in the cerebellum and associated structures, which would encourage its application in future clinical trials.

With respect to SCA3, the joint action of two shRNAs (shRatatax1 and shRatatax2) led to knock-down expression of *ATXN3* mRNA and protein in 293T cells [108]. Injection of the same mixture of shRNAs in a rat model of Machado–Joseph disease (MJD) decreased ataxin-3 levels and ataxin-3 inclusions at two months post-injection [108]. Furthermore, experiments in a humanized SCA3 mouse model, using an AAV-mediated delivery of small interfering RNAs (siRNAs), which targeted the 3′ UTR of *ATXN3*, led to gene silencing of human mutant *ATXN3* [109,110]. Interestingly, this therapeutic intervention prevented the nuclear accumulation of mutant ataxin-3 throughout the cerebellum [109]. Nonetheless, long-term treatment failed to ameliorate either motor impairment or short-term survival [110]. On the other hand, a life-long treatment was carried out in MJD mice of 6–8 weeks old, by injecting into cerebellar nuclei an AAV encoding microRNA-like molecules directed to the *ATXN3* 3′UTR [111] (Figure 3A). Such treatment effectively suppressed mutant *ATXN3* expression in the Purkinje cells and the deep cerebellar nuclei neurons for up to 9–10 months post-injection but failed to improve motor function and survival [111]. Likewise, the human *ATXN3* mRNA and protein levels were decreased by treatment with the ASO miR-Atx3-148, which was delivered into the cerebellum of MJD mice, via AAV-mediated delivery [109]. Furthermore, the expression levels of some microRNAs (miRNAs) were restored to those observed in untreated mice [109]. In addition, mitigation of behavior deficiency and neuropathology was obtained in SCA3 mice after siRNA siMutAtax3 administration, which effectively repressed mutant *ATXN3* expression [112]. Interestingly, the expression of four different miRNAs, namely, miR-25, miR-125b, miR-29a and miR-34b, was found to be altered in *ATXN3* patients [113], which suggests their involvement in *ATXN3* gene expression. Supporting this idea, transfection of miR-25 into SCA3 cells decreased mutant ataxin-3 protein aggregates and alleviated polyQ-associated cytotoxicity at 48 h post-transfection [36]. Notably, overexpression of three miRNAs that target the 3′UTR *ATXN3* (hsa-mir-9-5p, hsa-mir-181a-5p and hsa-mir-494-3p) had a therapeutic effect in 5-week-old SCA3 mice, as shown by the marked reduction in mutant ataxin-3 protein inclusions and the alleviation of neuronal dysfunction attained after their co-injection in the striatum [33].

Regarding SCA6, viral delivery of miR-3191-5p prevented motor deficiency and Purkinje cell impairment in an early-onset mouse model [114]. Kubodera et al. suppressed both WT and mutant alleles of the SCA6 causative gene (*CACNA1A*) in 293T cells, using a non-allele-specific siRNA [115]. Then, the expression of the WT allele was rescued using a second vector encoding an siRNA-resistant *ATXN6* cDNA [115]. A major limitation of this strategy is the difficulty of achieving the required WT ataxin-6 protein levels upon re-expression.

Concerning SCA7, rescue of the SCA7 phenotype in a mouse model was found upon bi-lateral injection into the deep cerebellar nuclei and further AAV-mediated expression of an siRNA [116]. In addition, these authors observed a decrease in the mutant *ATXN7* mRNA and protein levels and reduced thickness of the cerebellar molecular layer as a response of the treatment. In the same direction, in situ administration of this siRNA in the mouse retina, via subretinal injection of the corresponding AAV, resulted in a sustained (23 weeks) decrease in mutant *ATXN7* mRNA and protein levels, with no apparent toxicity [117]. An alternative approach to SCA7 therapy implicates the use of siRNAs composed of self-complementary CUG repeats, containing a single base mutation to facilitate guide strand self-duplex (sd) formation [118,119]. These sd-siRNAs efficiently formed base-mismatched complexes with their complementary CAG repeats, which in turn resulted in the mutant ataxin-7 protein silencing in SCA7 patient fibroblasts, through translation blockage [120]. The non-allele-specific silencing strategy has also been applied to SCA7. Simultaneous expression of artificial mirtrons against *ATXN7* mRNA and a functional mirtron-resistant *ATXN7* wild-type copy was successfully accomplished in patient-derived fibroblasts [121]. Mirtrons are introns that form pre-microRNA hairpins upon splicing. This approach can be used to silence mutant *ATXN7* expression and preserve normal protein function at the same time.

### 4.5. Allele-Specific siRNA-Mediated Gene Silencing

Allele-specific silencing using an SNP to discriminate between WT and mutant transcripts is a promissory strategy against polyQ SCA disorders [122,123] (Figure 3B). Allele-specific inhibition of mutant ATN1 protein expression was obtained in DRPLA patient-derived fibroblasts, using RNA duplexes that contain mismatched bases respective to the CAG target [124]. Additionally, identifying ATXN7 gene-linked SNPs has facilitated the design of allele-specific silencing strategies for this ataxia [125,126]. Remarkably, specific suppression of mutant *ATXN7* transcripts and decreased mutant protein aggregates, with no effect on wild-type mRNA, were achieved in patient-derived fibroblasts using shRNAs that target SNPs [127]. Unfortunately, this approach applies only to the subset of patients that carry a targetable SNP.

### 4.6. Gene Editing

The emergence of genome editing technology, including clustered regularly interspaced short palindromic repeats (CRISPR)-associated protein 9 (Cas9), transcription activator-like effector nucleases (TALENs) and zinc-finger nuclease (ZFN) platforms, has allowed targeting and editing polyQ-disease genes [92,128,129] (Figure 3D–F). An allele-specific CRISPR/Cas9-mediated strategy was successfully applied to HD [130]. Likewise, CRISPR/Cas9-based deletion of the *ATXN3* gene exon 10, which carries the expanded CUG tract, was successfully obtained in patient-derived iPSCs [131]. Remarkably, genetically modified *ATXN3* did not form aggregates but maintained its ubiquitin binding activity; furthermore, iPSCs preserved their capacity for differentiation. Despite the high expectations to apply gene editing technology in the clinic, its potential off-target effects that might provoke undesired genomic rearrangements are still a major concern [132].

## 5. Limitations of Gene Therapy

Although different oligonucleotide technologies aimed to induce degradation or neutralization of toxic RNA or to edit disease-causing mutations have provided promising results in cellular and/or animal models, several hurdles are required to be overcome before their implementation in clinical trials. For instance, ASO-based approaches require continuous re-administration of ASOs to offer long-term alleviation. Thus, the development of more stable nucleic acid chemistries is needed to reduce the dosing frequency. Furthermore, the BBB integrity significantly limits the uptake of ASOs, which ultimately causes their poor bioavailability and biodegradation [133,134]. Favorably, conjugates of morpholino ASOs with arginine-rich cell-penetrating peptides or siRNA with the antigen-binding fragment (Fab) confer to them the ability to penetrate cellular membranes, including the BBB [135,136]. Specifically, chimeric peptides carrying both brain-specific heptapeptides and cell-penetrating peptide domains effectively mediated the delivery of morpholino AON and siRNAs to the CNS [137,138]. Nonetheless, inefficient processing of shRNAs into mature siRNA can provoke neurotoxicity. In addition, high concentrations of siRNA can saturate the RNAi system, leading to a global perturbation of miRNA-mediated regulation [139,140]. Finally, as mentioned above, most ASO-mediated strategies recognize and induce the cleavage of both wild-type and mutant mRNA alleles with similar efficacy, and the design of allele-specific approaches is greatly limited by the low probability of finding an SNP within the mutant allele sequence.

On the other hand, virus-based vectors, including lentivirus and adeno-associated virus, provide highly effective and long-term production of therapeutic molecules; however, high-pressure intravascular delivery is required for drugs to reach the different brain regions. In addition, the immune system response to viral proteins possesses a major obstacle [141]. In this scenario, the use of nanotechnology to carry and efficiently deliver therapeutic molecules to the brain is an excellent alternative for treating polyQ SCA diseases [85,142].

## 6. Novel Nanovector Tools for Brain Delivery of Therapeutic Molecules

Current therapeutic vectors have physicochemical characteristics that limit their ability to pass through cellular barriers to reach the brain. To improve the brain bioavailability of therapeutic drugs, new formulations based on nanocarriers emerge. A well-designed nanocarrier system must meet several requirements, including high stability and specificity, proper tissue distribution, effective cell penetration and efficient cytoplasmic or nuclear delivery [143]. The physicochemical properties of nanocarriers including size, shape, surface charge, porosity and crystalline arrangement determine their physiological behavior. As broad types of nanoparticles (NPs) made of biological and/or synthetic materials are available, selecting suitable particles relies mainly on the type of therapeutic molecule to be transported.

Although viral vectors exhibited high transfection efficiency, their mutagenic/oncogenic potential, as well as their limited drug loading capacity and high-cost production, has discouraged their use [143,144]. Conversely, NPs constitute a suitable carrier system because they show high biocompatibility and evoke virtually no immune response. Moreover, NPs provide cargoes protection against chemical and biological degradation, and their synthesis is comparatively cheaper than that of viral vectors. Additional advantages of NPs include their ability to penetrate deep tissues, tiny capillaries and cell membranes [144,145].

NP-based carrier systems designed for the brain delivery of therapeutic drugs and molecular tools are composed of different materials, including polymers (nanocapsule, nanosphere, polyplex and nanogel), lipids (solid lipid nanoparticle, nanoliposome, lipoplex and polymersome), metals (magnetic nanoparticle, gold nanoparticle and silica nanoparticle) or a combination of two of the above-mentioned materials (Figure 4). In addition, nanoparticle formulation must provide high biocompatibility, biodegradability, no toxicity and a low level of protein-mediated opsonization [145,146]. In addition, the undesired action of NPs on platelet activation and their clearance by the reticuloendothelial system are some obstacles for NPs to overcome [145,146].

### 6.1. Lipid-Based Nanoparticles

Lipid-based NPs are commonly used as carriers for biological molecules in gene therapy. Lipids allow the configuration of different NP systems, including micelles (small unilamellar vesicles with a hydrophobic core and hydrophilic shell), solid lipid NPs (SLNs; micellar vesicles with a hydrophobic solid lipid core that prevents lipid permeation and degradation) and liposomes (large vesicles containing a lipid bilayer that forms a hydrophilic core and shell) (Figure 4) [147]. Different lipid-based NPs have been employed in gene therapy. For example, a liposomal system composed of nucleic acid lipid particles, which incorporated a short peptide derived from the rabies virus glycoprotein and embedded siRNAs against mutant *ATXN3* mRNA, was evaluated in two mouse models of MJD [112]. Upon intravenous administration of these liposomes, the mutant ataxin-3 expression was effectively repressed, which in turn ameliorated motor behavior and neuropathological alterations, showing the ability of liposomes to carry and preserve siRNAs’ functionality in the brain parenchyma [112]. Likewise, NPs composed of hydrogenated soya phosphatidylcholine 3-nitropropionic acid and Tween 80 were synthesized to deliver the neuroprotector rosmarinic acid to the brain of Wistar rats, previously treated with 3-nitropropionic acid (3-NP) to induce HD-like symptoms. After nasal administration of NPs, attenuation of 3-NP-induced behavioral abnormalities and oxidative stress were found in these rats [148]. Similarly, Rassu G. et al. designed chitosan-coated and uncoated SLNs to achieve a nose-to-brain transport of BACE1 siRNA [149]. The authors found that this nanoformulation increased the permeation of the BACE1 siRNA through the epithelial monolayer of Caco-2 cells at early times upon administration (1 h), as suggested by the chitosan action. Furthermore, two NP systems, polymer (poly (lactic-co-glycolic acid), PLGA) and solid lipid (Witepsol E85) NPs, were functionalized with a peptide-binding transferrin receptor to improve their ability to target human brain endothelial cells and deliver siRNAs, with no significant toxicity [150].

### 6.2. Polymeric Nanoparticles

Polymeric NPs possess several properties for precise drug delivery to the CNS, including their biocompatibility and biodegradability. The most common methods for polymeric nanoparticle synthesis are nanoprecipitation, emulsification–diffusion, emulsification–solvent displacement and salting-out [142]. Hydrophobic polymers and copolymers such as poly L-lysine (PLL), polyethyleneimine (PEI), poly (lactic acid) (PLA), PLGA and poly(ε-caprolactone) (PCL), as well as hydrophilic polymers such as chitosan, alginate, gelatin and hyaluronic acid, are commonly used for NPs’ preparation. These polymers, alone or in combination, can entrap biomolecules of interest for molecular therapies. In this context, it has been shown that PLGA NPs improve drug stability and maintain a sustained release of DNA, ASOs and siRNA [151,152]. Specifically, PLGA can react as a cationic condenser and thereby promotes entrapment of anionic DNA and similar molecules [147]. Recently, a PLGA nanocarrier was synthesized to deliver MDR-1 and BCL2 siRNAs, with the aim of simultaneously suppressing the drug efflux and anti-apoptotic pathways, which are involved in multidrug-resistant ovarian cancer cells [153]. Likewise, brain distribution of aripiprazole (APZ; a small molecule that reduces the levels of mutant ataxin-3 protein [47,154]) was facilitated when it was loaded into PCL NPs and intranasally administered to rats [155]. These PCL NPs exhibited a particle size of 199.2 ± 5.65 nm, a zeta potential of −21.4 ± 4.6 mV and an APZ EE of 69.2 ± 2.34%. Regarding cellular models, PCL/F68 NPs synthesized by emulsification–diffusion were effective in mediating the delivery of curcumin (antioxidant compound) to neural-like cells with low cytotoxicity [156]. On the other hand, NPs based on natural polymers, such as chitosan and cyclodextrins, have been found to be suitable delivery systems because of their ability to cross the blood–brain barrier [157,158]. In this context, chitosan/PLA/polyethylene glycol (PEG) NPs complexed with the nerve growth factor acteoside and plasmid DNA (pDNA) were developed to treat Parkinson’s disease [158]. Interestingly, this complex formulation inhibited the expression of alpha-synuclein upon internalization into PC12 cells. With respect to cyclodextrins, an siRNA loaded with amphiphilic β-cyclodextrins was designed for Huntington’s disease treatment [159]. The authors observed a remarkable reduction in the toxic Huntingtin mRNA levels, with low cytotoxicity.

### 6.3. Polyplexes

Polyplexes are polymeric systems assembled through electrostatic interactions between the cationic polymer groups and the negatively charged nucleic acids. Polyplexes require the ability to internalize the cell and escape from endosomes to reach an effective delivery of drug or nuclei acids [160,161]. In a recent work, a poly(trehalose) formulation was designed to deliver siRNAs in glioblastoma cells, since this polymer possesses the cryoprotection capability of trehalose, while maintaining the siRNA biological role [162]. These authors reported effective internalization of poly(trehalose) and further downregulation of siRNA-targeted genes [162]. Likewise, arginine-rich polyplexes modified with short-chain PEG showed high internalization of pDNA into U-87 cells, including the nucleus [163]. In 2020, Koji, K et al. developed polyethylene glycol (PEG)-coated polyplex micelles containing highly condensed mRNA. Interestingly, these loaded bundled mRNA micelles showed high stability in mouse blood flow and evoked efficient green fluorescent protein expression in both cultured cells and mice brain tissues [164].

### 6.4. Metallic Nanoparticles

Metallic NPs have been proposed as alternative therapeutic carriers in biomedicine because of their unique physiochemical properties. Supporting this idea, aggregation of polyQ-containing mutant Huntingtin was hampered in neuronal cells and HD mouse brains, by using an Fe_2_O_3_ polyacrylate-coated and covalently conjugated poly(trehalose) nanocarrier system [165]. The authors reported a nanosystem with a size of 20–30 nm. Thus, a similar metallic NP-based approach could be scaled to treat polyQ SCAs. However, the fact that metallic NPs are not biodegradable and therefore accumulate in neuronal cells could cause toxicity.

## 7. Targeting Strategies for Delivery of NPs into the Brain

Adjustment of physicochemical properties, including nanoparticle size, morphology, electric charge density on the surface and capability of protection, can improve the ability of nanocarriers to transport therapeutic cargoes to the brain. The delivery of therapeutic molecules into the brain parenchyma largely depends on the carrier’s capacity to cross through the BBB, a very specialized structure formed by brain capillary endothelial cells and sustained by astrocytes and pericytes. Transportation of drugs across the BBB involves different routes (see below and Figure 5). The design of multifunctional NPs with decorated surfaces confers the ability to pass through the BBB, and NPs with modified surfaces can cross the BBB using passive or active targeting processes, as described in the following sections.

### 7.1. Passive Targeting

Passive targeting of NP systems does not require targeting ligands; instead, it needs enhanced permeability and a retention (EPR) effect. The critical parameters for NPs to exert passive targeting include the EPR effect, a proper size and surface charge, an appropriate zeta potential and lifetime in the blood circulation. Some of these parameters, such as zeta potential, lifetime and electric charge density, can be modulated using a flexible, hydrophilic polymer coating. For instance, PEG NPs with charged surfaces acquired a steric barrier that hinders their interaction with blood components; consequently, the adsorption of plasma proteins on their surface is limited, thereby preventing rapid opsonization of NPs by the matrix metalloproteinases (MPS). Thus, an increased circulation time in the blood of NPs offers a better chance of extravasation through vascular tissue [166,167]. In this regard, cationic bovine serum albumin (CBSA) NPs, conjugated with PEG and PLA, were efficiently uptaken by rat brain capillary endothelial cells (BCECs) upon incubation at 37 °C. Furthermore, CBSA NPs accumulated at a high concentration in different brain coronal regions upon injection in the mouse caudal vein [168]. Likewise, two nanosystems termed CBSA NPs and BSA NPs were manufactured for passive targeting [169]. These NPs were administered to Sprague Dawley rats via the tail vein, and a fluorescence probe was coupled to NPs to track their trafficking to the brain. Interestingly, improved accumulation of CBSA NPs in the brain was found, compared with BSA NPs and regular CBSA conjugation, which implies that this nanosystem enhanced bioretention [169].

Besides a prolonged lifetime in the blood, NPs need to release biomolecules into the cell, where the lysosomal pathway might eliminate the therapeutic drugs. The sponge hypothesis postulates how nucleic acids/drugs can escape from endosomes. This hypothesis is based on the gradual osmotic change that leads to vesicle swelling and rupture to release the loaded molecules [151]. Polymer-like polyethyleneimine (PEI) contains abundant nitrogen groups to buffer pH and sponge up protons. Nonetheless, PEI-mediated sponge mechanisms remain to be demonstrated [147]. In this regard, the poly(ethylenimine)-cholesterol (PEI-Chol) nanosystem exhibited a high transfection efficiency with little toxicity in Jurkat cells, as shown by green fluorescent protein expression, implying efficient endosomal release [170]. It is thought that NPs coated with polymers improve their ability to deliver drugs to the brain parenchyma. Supporting this assumption, poly (butyl cyanoacrylate) NPs loaded with dalargin or loperamide and coated with polysorbate 80 and apolipoprotein B or E evoked an antinociceptive effect in mice models upon intravenous injection [171]. The authors claimed that uptake of polysorbate 80-coated NPs by the brain capillary endothelial cells occurred via receptor-mediated endocytosis. Likewise, albumin-lipid NPs with entrapped docetaxel (a chemotherapy drug) accumulated in the brains of glioma-bearing mice at 4 h post-injection, which suggests NPs’ associated passive targeting and the EPR effect [172].

### 7.2. Active Targeting

Active targeting facilitates the uptake of NPs by the cells themselves. This process is classified into three subcategories: transporter-mediated transcytosis, adsorptive-mediated transcytosis and receptor-mediated transcytosis [173,174] (Figure 5). Adsorptive-mediated transcytosis is a nonspecific process that involves the interaction of positively charged peptides or proteins with negatively charged microdomains on the membrane of brain endothelial cells. Transporter-mediated transcytosis is a substrate-selective transport that implicates the internalization of small biomolecules. It is applied to deliver nutrients such as glucose, amino acids and purine bases to the brain. Receptor-mediated transcytosis is a highly selective and specific transport related to the binding between targeting ligands and receptors expressed on the brain endothelial cells [175,176]. Due to the controlled release capacity, protection of macromolecules from degradation and unneeded efflux, receptor-mediated transport is one of the most promising brain drug delivery strategies.

Thus, surface modifications to improve NPs’ crossing of the BBB and their uptake by specific cells using different ligands have been approached, including transferrin (Tf) receptor, lactoferrin (Lf) receptor, insulin receptor (IR), low-density lipoprotein receptor-related protein (LDLR) and nicotinic acetylcholine receptors (nAChRs). Table 4 illustrates the different transport systems and some examples of their applications on brain-targeting delivery.

## 8. Transporter-Mediated Strategies for Drug Delivery into the Brain

### 8.1. Glucose

Since glucose is a key energy source for various biochemical reactions, the cell possesses a variety of glucose transporters [190,191]. Therefore, glucose-modified NPs have been evaluated to enhance the efficacy of therapeutic drug delivery. Hydroxyapatite NPs with or without glucose modification were complexed with plasmid DNA and further transfected into endothelial cells [192]. The authors observed increased internalization of glucose-modified HAp/pDNA NPs, compared with unmodified NPs, and demonstrated that the uptake route was the glucose transporter 1 (GLT1) [192].

### 8.2. Transferrin

Transferrin (Tf) is a monomeric glycoprotein consisting of two homologous lobes called N and C lobes, which are connected to each other by a short peptide. The transport of plasma circulating iron into BBB endothelial cells via transferring receptor (TfR) is a classic example of receptor-mediated endocytosis (Figure 6) [175,193,194,195]. In this context, the therapeutic effects of an siRNA delivery system against *EGFR* were evaluated in vivo, composed of Tf-mediated core-shell NPs (T7-LPC/siEGFR NPs) [196]. Remarkably, downregulation of EGFR expression in tumor tissues was found in mice bearing an intracranial U87 glioma treated with T7-LPC/siEGFR NPs, compared with mice treated with other formulations [196]. Since apolipoprotein E2 (ApoE2) is employed in Alzheimer’s disease (AD) gene therapy, the therapeutic effect of liposomes entrapping ApoE2 plasmid DNA, which were functionalized with transferrin and penetratin (Tf-Pen-liposomes), was analyzed. Interestingly, increased ApoE2 levels were observed in mouse brains after intravenous administration of Tf-Pen-liposomes [197].

### 8.3. Lactoferrin

Lactoferrin (Lf) is an iron-carrying glycoprotein that belongs to the Tf protein family. Lf’s polypeptide chain (approximately 690 residues) is folded into two globular lobes connected by a short peptide forming a three-turn α-helix. It is synthetized by mucosal epithelial cells and neutrophils in various mammalian species and is involved in various protective activities related to antioxidant, anticancer, anti-inflammatory and antimicrobial activities [198,199]. Experimental evidence demonstrated that brain capillary endothelial cells internalized Lf-modified NPs efficiently [195]. Furthermore, the neuroprotective effect of the human *GDNF* gene (hGDNF) was analyzed in a rotenone-induced chronic Parkinson’s disease model, using hGDNF-loaded NPs modified with Lf [199]. Supporting the role of Lf as a ligand to improve the BBB crossing of therapeutic agents, the locomotor activity of rats was improved upon multiple intravenous injections of Lf-modified NPs loaded with hGDNF, with a reduction in the loss of dopaminergic neurons and an increase in monoamine neurotransmitter levels [199]. In a subsequent study, a higher accumulation of Lf-modified NPs in the brain was confirmed by in vivo imaging, compared with unmodified NPs [200]. Owing to the presence of Lf receptors in the brain, the use of Lf-modified NPs could be adapted to treat different polyQ-disorders. In a recent study, higher cellular uptake of Lf-conjugated PEG-PCL NPs was observed in brain endothelial cells than Lf-free NPs [201]. Likewise, higher accumulations of PLGA- and PEG-conjugated Lf-NPs were found in the mouse brain after intravenous administration, compared to unconjugated NPs [202]. In line with the above data, the anti-neoplastic effect of temozolomide (TMZ) was enhanced by using TMZ-loaded NPs functionalized with Lf (TMZ-LfNPs) as the delivery system for the brain [198]. These Lf NPs could cross the BBB and target overexpressed Lf receptors on mouse glioma to further release TMZ, as shown by the significant reduction in tumor volume and improved median survival of glioma-bearing mice [198]. Finally, Qingqing Meng et al. found high concentrations of Lf-modified NPs in different mouse brain regions after intranasal administration, including the mouse olfactory bulb, cerebellum and hippocampus [203]. Overall, these results demonstrate that Lf-modified NPs present an efficient brain delivery platform for therapeutic drugs.

### 8.4. Insulin Receptor

The insulin receptor (IR) is another receptor expressed at the BBB, which transports insulin to the brain in a receptor-mediated transport fashion. However, insulin has barely been used in nanosystems, due to its short serum half-life (10 min), and the possibility of triggering hypoglycemia. Instead, an anti-IR monoclonal antibody-based strategy (mAbs) has been tested for brain delivery of drugs [204]. In this regard, carmustine (BCNU)-loaded solid lipid NPs (SLNs) attached to anti-insulin monoclonal antibody 83-14 (83-14 MAb/BCNU-SLNs) were proved for brain targeting [205]. Noteworthy, 83-14 MAb/BCNU-SLNs promoted endocytosis in vitro in human brain endothelial cells via IRs and enhanced BCNU permeability through the BBB. Finally, insulin or anti-insulin receptor monoclonal antibodies (29B4) covalently coupled to human serum albumin (HSA) NPs were able to transport loperamide across the BBB, as shown by the induction of antinociceptive effects in ICR (CD-1) mice after intravenous injection [206].

### 8.5. Low-Density Lipoprotein Receptor-Related Protein

Apolipoprotein E (ApoE) is a structural component of lipoproteins, acting as a ligand to bind specific cell surface receptors and lipid transport proteins. ApoE maintains cholesterol homeostasis by binding to specific cell surface receptors, including low-density lipoprotein receptor (LDLR) and LDLR-related protein 1 (LRP1) [207] (Figure 6). Therefore, ApoE is another attractive molecule to improve nanocarriers’ transport through the BBB. Ana Rute Neves et al. functionalized resveratrol-loaded SLNs with ApoE to enhance BBB permeability via LDL receptors [208]. The authors observed a significant increase in drug permeability in hCMEC/D3 cells when ApoE-SNLs were applied compared to non-functionalized SLNs. Furthermore, a delivery system composed of ApoE-modified liposomal NPs was found to stimulate the uptake of siRNAs by brain endothelial cells in an ApoE concentration-dependent manner [209]. They also demonstrated that the internalization of ApoE-modified NPs occurred through both clathrin-mediated endocytosis and caveolae-mediated endocytosis.

## 9. Proof of Concept of NPs Delivery on PolyQ Diseases

In recent studies, active and passive targeting of NPs has been approached in polyQ disorder models, such as Huntington’s disease. In this regard, a system comprising PLGA NPs loaded with synthetic peptides (QBP1 and NT_17_) and coated with polysorbate 80 inhibited polyQ protein aggregation in both Neuro 2A and PC12 cells [210]. Moreover, the larval crawling activity was significantly higher in an HD *Drosophila* model upon dosages of peptide-loaded polysorbate 80-coated NPs, compared to empty NPs [210]. Likewise, biodegradable trehalose-conjugated catechin-loaded polylactide NPs were elaborated to enhance neuroprotection against polyQ expansion in Huntingtin [211]. Trehalose possesses the ability to impede protein aggregation [165]. Strikingly, hindered polyQ aggregation decreased oxidative stress, and an augmented proliferation was observed in HD150Q cells upon NP uptake. In a recent work, LNP-loaded siRNA duplexes, which target CAG repeats, were able to suppress the polyQ-expanded androgen receptor (AR) expression in cultured cells [212]. Remarkably, the LNP-mediated delivery system was effective in selectively suppressing the mutant AR in both the central nervous system and the skeletal muscle of an SBMA mouse model [212].

## 10. The Challenge of Cytoplasmic Transit and Nuclear Internalization of NPs

The entry of NPs into neurons is influenced by their size and the interaction between surface components and the cell membrane (passive and active targeting, see previous sections). Typically, the smaller the size, the greater the capacity of internalization into the cytoplasm. In addition, the surface composition and architecture can favor and complement the internalization capacity of NPs [213] (Figure 6). Internalization strategies involve the presence of the stabilizer type, the zeta potential and the presence of cell-penetrating peptides (CPPs) on the surface [214,215]. The most widely used CPPs for NPs’ docking is the transactivating-transduction (TAT) peptide (GRKKRRQRRRPQ), which is derived from human immunodeficiency virus. CPPs can be categorized into three types: amphipathic, cationic and hydrophobic. Amphipathic CPPs are internalized by macropinocytosis through an adsorption-governed process due to lipophilic–hydrophilic interactions on the cell membrane surface. Examples of this type of CPP include Pep-1, MPG, pVEC, MAP and CADY. Cationic CPPs generated an opposite charge-based assembly on the cell membrane via the interaction of negatively charged phosphates with sulfates, thereby facilitating internalization; examples of these CPPs include polyarginine, TAT 49-57, penetratin, P22N, DPV3 and DPV6CADY. Finally, hydrophobic CPPs contain hydrophobic amino acids that form hydrogen bonds and favor their passage through the cell membrane. Hydrophobic CPPs include K-FGF and C105Y [214]. Typically, NPs suffer encapsulation inside endosomes after internalization, followed by their fusion with lysosomes (Figure 6) [216]. Endosome-mediated encapsulation is accompanied by acidification, with the presence of proteases, lipases and nucleases in these structures. As NPs can suffer degradation via lysosomes, some approaches have been designed to escape the endo-lysosomal degradation pathway. For instance, “proton sponge” strategies include the use of PEI or low pH-sensitive fusogenic peptides derived from the amino-terminus sequence of influenza virus hemagglutinin 2, in order to destabilize the endosomal membrane and provoke NP release [213]. Besides the escape from lysosomes, NPs need to enter the nucleus to accomplish their function.

L-lysine repeats promote nuclear accumulation of NPs after caveolin-mediated endocytosis and evasion of lysosomal degradation by trafficking through the endoplasmic reticulum (ER) and Golgi apparatus (GA). The above route is safe for various NPs. Likewise, _L_-arginine and saccharide moieties evoke the same caveolae-mediated nucleus targeting pathway, using membrane transit through the ER and GA. Another nuclear internalization strategy involves the passage through the nuclear pore complexes (NPCs) by using importin-mediated active transport governed by nuclear localization signals (NLSs), which are short amino acid (aa) sequences (7–10 aa) rich in basic residues [216,217] (Figure 6). Some NLSs that have been coupled to NPs include the sequences CGGGPKKKRKVGG, CGYGPKKKRKVGG and H-Cys-Gly-Gly-Arg-Lys-Lys-Arg-Arg-Gln-Arg-Arg-Arg-Ala-Pro-OH [213,216].

## 11. Conclusions

Recent advances in understanding the molecular basis that underlies polyQ SCAs have allowed the development of ASOs/siRNA-based and gene editing strategies to silence the mutant RNA expression and correct the causative disease DNA mutations, respectively. Despite the tremendous progress obtained by evaluating gene therapy approaches in different cellular and animal models, their implementation in clinical trials is limited due to their toxicity, poor BBB permeability and/or metabolic instability. NP systems offer an attractive and feasible strategy to improve drug delivery to the brain because of their flexibility to encapsulate different therapeutic molecules and cross the BBB and deliver their content in a prolonged/modulated manner. Future development of NPs with suitable physicochemical and functionalization characteristics will surely facilitate the translation of such technological findings to clinical trials for treating polyQ SCAs.

## Figures and Tables

**Figure 1 pharmaceutics-13-01018-f001:**
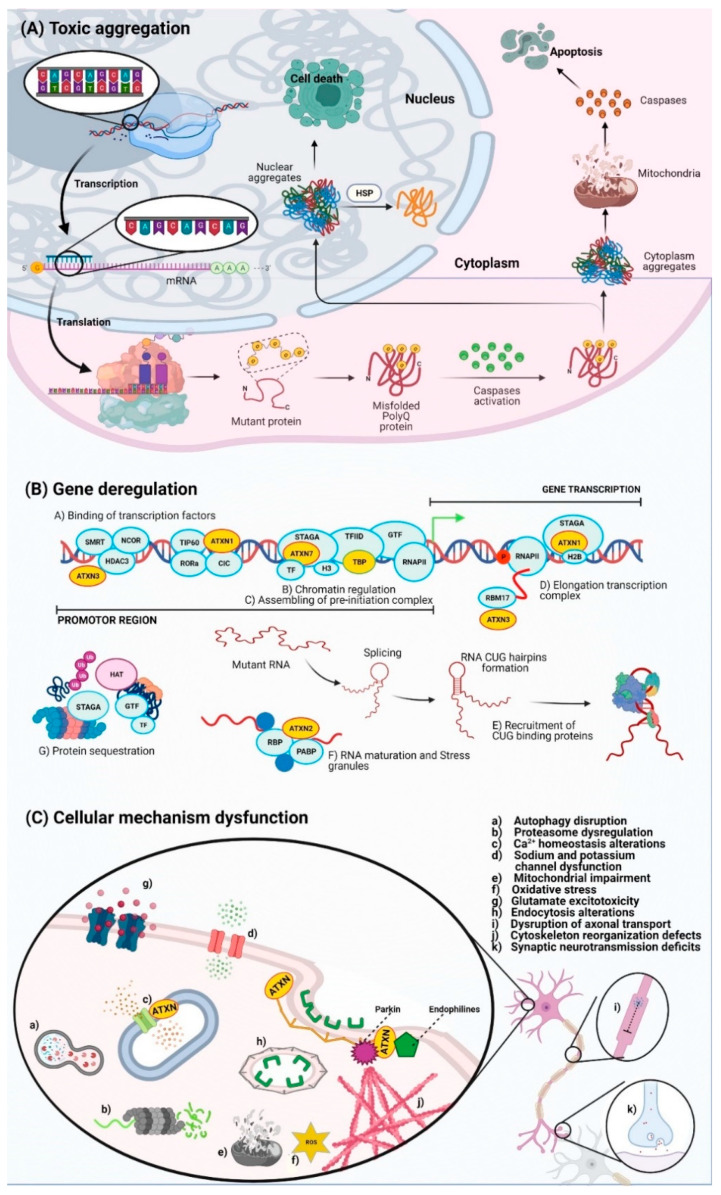
PolyQ protein aggregates disturb different cellular processes. (**A**) Intracellular fate and aberrant aggregation of polyQ proteins. (**B**) Role of mutant proteins and their corresponding RNA transcripts in gene expression. (**C**) Altered cellular pathways that trigger neurodegeneration in polyQ SCAs. See the text for details.

**Figure 2 pharmaceutics-13-01018-f002:**
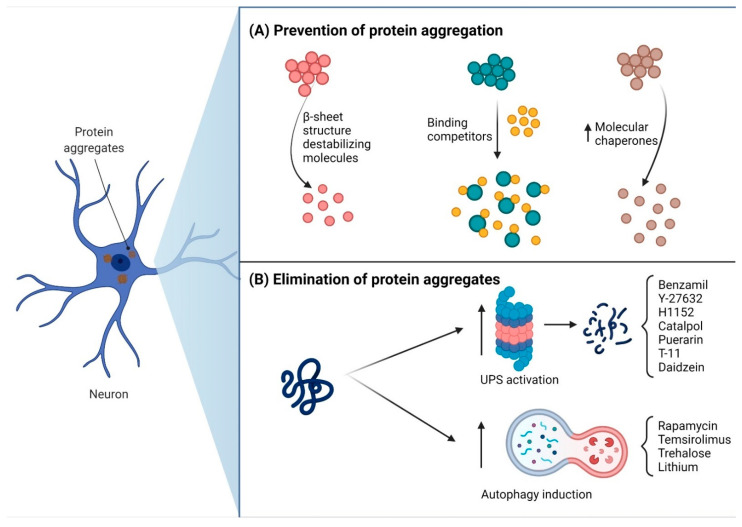
Pharmacological therapeutic perspectives for SCA. (**A**) Three different strategies to prevent mutant protein aggregation. (**B**) Elimination of protein aggregates by the enhancement of ubiquitin–proteasome system (UPS) or autophagy with different drugs.

**Figure 3 pharmaceutics-13-01018-f003:**
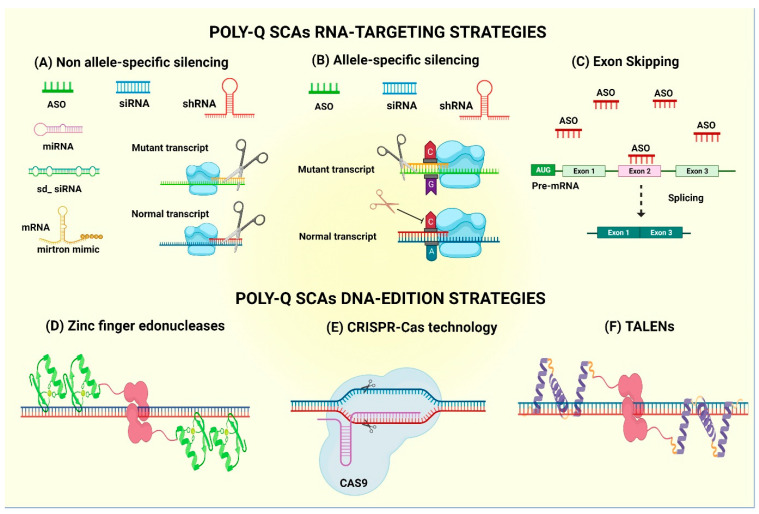
Molecular strategies for polyQ SCA therapy. (**A**) Non-allele-specific silencing of mutant genes by using antisense oligonucleotides (ASOs), small interfering RNAs (siRNAs), short hairpin RNAs (shRNAs), microRNAs (miRNAs), sd-siRNA or mirtrons. (**B**) Allele-specific silencing using ASOs or shRNAs that target a single-nucleotide polymorphism (SNP) linked specifically to the mutant allele, in order to induce its cleavage while maintaining intact WT mRNA expression. (**C**) Exon skipping on primary mutant RNAs by splicing blockage with ASOs. (**D**) Gene editing using zinc-finger endonucleases. (**E**) CRISPR-Cas9 technology. (**F**) Gene editing by TALENs.

**Figure 4 pharmaceutics-13-01018-f004:**
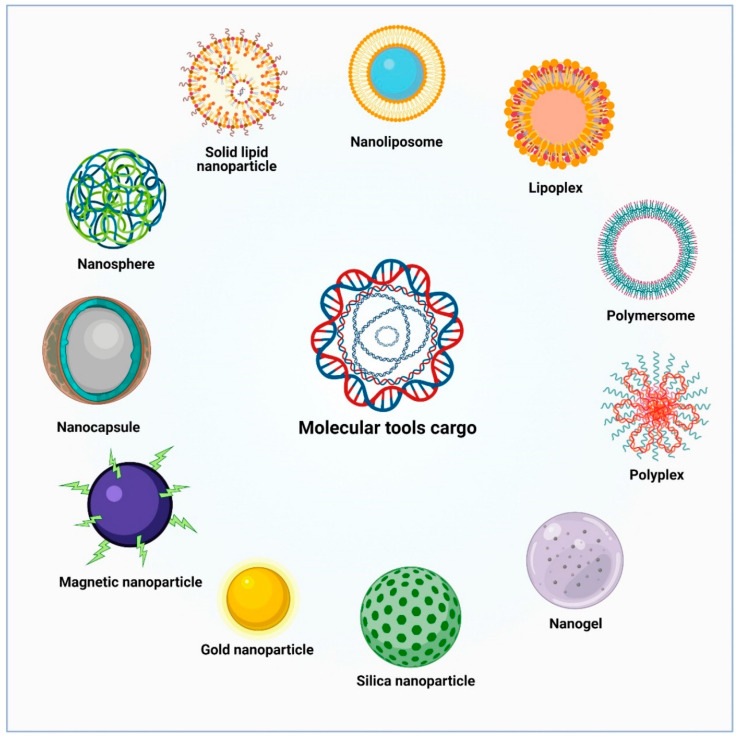
The scheme shows types and morphologies of the NP systems used for treating polyQ SCA diseases.

**Figure 5 pharmaceutics-13-01018-f005:**
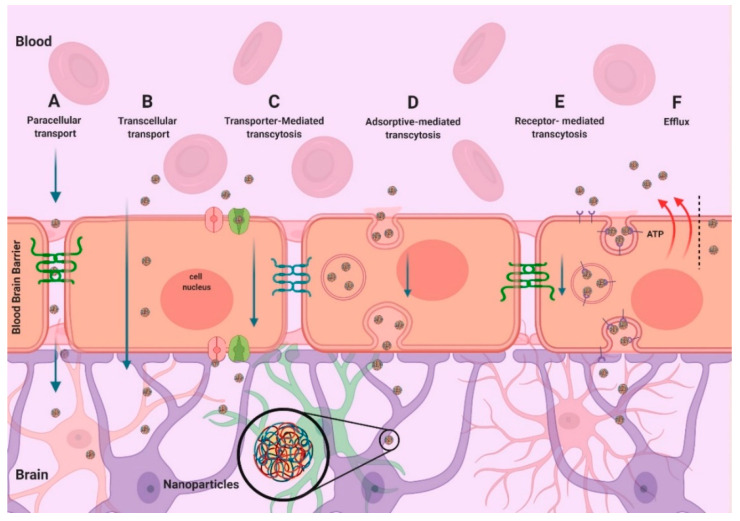
Mechanisms for crossing the BBB: (**A**) paracellular transport, (**B**) transcellular transport, (**C**) transporter-mediated transcytosis, (**D**) adsorptive-mediated transcytosis, (**E**) receptor-mediated transcytosis, (**F**) efflux. The mechanisms indicated in (**C**–**E**) have been explored to overcome the BBB restrictiveness and enhance drug delivery into the brain parenchyma.

**Figure 6 pharmaceutics-13-01018-f006:**
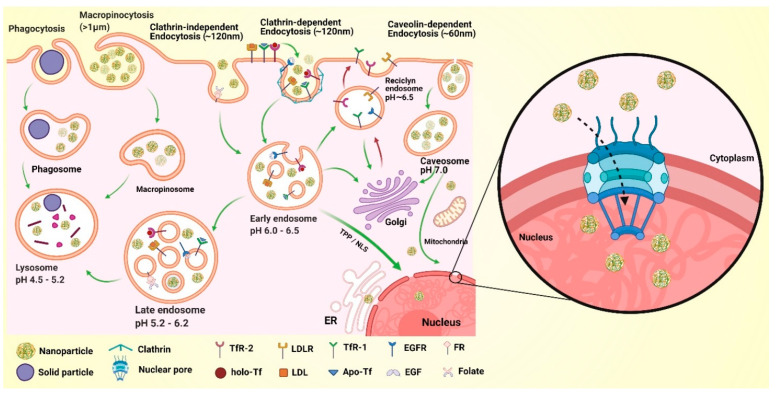
Receptor-mediated endocytosis. Macropinocytosis forms macropinosomes that in turn generate early endosomes. Clathrin-mediated endocytosis (CME) and caveolin-mediated endocytosis (CVME) are the main receptor-mediated endocytosis (RME) mechanisms. Additional endocytic mechanisms have been described, including those mediated by flotillin, ARF6, RhoA and CDC42. In neurologic tissues, CME is the predominant mechanism of endocytosis. The final fate of endosome and endosome-like vesicles is to fuse with lysosomes. Thus, endosome-containing NPs need to escape from lysosomes to deliver the molecular tools within the nucleus.

**Table 1 pharmaceutics-13-01018-t001:** Genetic and clinical pathogenesis of polyQ-spinocerebellar ataxias.

Disease	Gene	Mutation (Localization)	Normal Alleles	Full Penetration Alleles	OMIM	Clinical Features	Neuropathological Findings	Ref.
SCA1	ATXN1	(CAG)n Exon 8 6p22-23	6–39	>40	164400	Ataxia, slurred speech, spasticity, cognitive impairment	Atrophy of cerebellum, pons and olives. Degeneration of lower cranial nerve nuclei, and atrophy of the dorsal columns, and spinocerebellar tracts. Loss of Purkinje cells, neurons of dentate gyrus, Bergmann’s gliosis, mesencephalic neurons in 3rd and 4th cranial nerves, variable loss of granule cells, atrophy of middle cerebellar peduncles. Intranuclear inclusions.	[15,16]
SCA2	ATXN2	(CAG)n Exon 1 12q23-24.12	14–31	>34	183090	Ataxia, slow saccades, decreased reflexes, polyneuropathy, motor neuropathy, infantile variant	Atrophy of cerebellum, pons, frontal lobe, medulla oblongata, cranial nerves, as well as pallor of the midbrain substantia nigra. Cytoplasmic inclusions.	[17]
SCA3	ATXN3	(CAG)n Exon 10 14q32.1	12–44	>52	109150	Ataxia, parkinsonism, severe spasticity	Loss of neurons and gliosis in the substantia nigra, pontine nuclei, nuclei of the vestibular and cranial nerves, columns of Clarke and anterior horns. The cerebellum is relatively spared, spinal cord with loss of myelinated fibers in the spinocerebellar tracts and posterior funiculi. Intranuclear and cytoplasmic inclusions.	[18]
SCA6	CACNA1A	(CAG)n Exon 47 19p13	4–18	>19	183086	Ataxia, dysarthria, nystagmus, tremor	Selective atrophy of the cerebellum and extensive loss of PC in the cerebellar cortex. Numerous oval- or rod-shaped, not ubiquitinated aggregates are seen exclusively in the cytoplasm of PC.	[19]
SCA7	ATXN7	(CAG)n Exon 3 3p12-21.1	4–35	>47	164500	Ataxia, retinal degeneration	Neuronal intranuclear inclusions in multiple brain areas, although more frequent in the inferior olivary complex, the lateral geniculate body, the substantia nigra and the cerebral cortex. Olivopontocerebellar atrophy and thinning of the spinal cord. Retinal degeneration.	[20]
SCA17	TBP	(CAG)n Exon 3 6q27	29–42	>47	607136	Ataxia, pyramidal and extrapyramidal signs, cognitive impairment, dementia, psychosis, bradykinesia and seizures	Mild neuronal loss with compaction of the neuropil in the cerebral cortex, striatum and moderate loss of PC. Nuclear inclusions.	[21]
DRPLA	ATN1	(CAG)n 12p13.31	6–35	>49	125370	Ataxia, epilepsy, choreoathetosis, dementia	Atrophy and neuronal loss in the globus pallidus (particularly the lateral segment) and dentate nucleus, brainstem, cerebellar and cerebral white matter. Lipofuscin deposits. Nuclear and cytoplasmic inclusions.	[22]

ATXN: ataxin; CACNA1A: calcium voltage-gated channel subunit alpha1 A; TBP: TATA-binding protein; ATN: atrophin.

**Table 2 pharmaceutics-13-01018-t002:** Current pharmacological treatments for polyQ SCAs.

PolyQ Disease	Current Treatment	Molecular Structure	Ref.
SCA 1	4-aminopyridine (4-AP) to ameliorate motor coordination deficiency of mouse model.	4-AP: 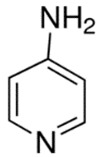	[52]
SCA 2	Levodopa to alleviate rigidity/ bradykinesia. Magnesium, quinine, mexiletine or vitamin B to ameliorate painful muscle contractions. Chlorzoxazone and riluzole (potassium channel modulators) to improve cerebellar electrophysiology.	Levodopa: 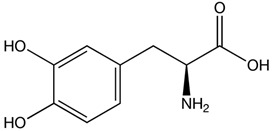 Chlorzoxazone: 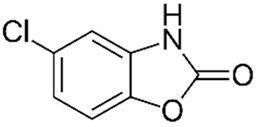 Riluzole: 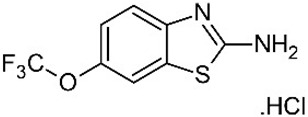	[17,41,46,48,49,51]
SCA 3	Varenicline (a partial agonist at α4β2 neuronal nicotinic acetylcholine receptors) to improve axial symptoms and rapid alternating movements.	Varenicline: 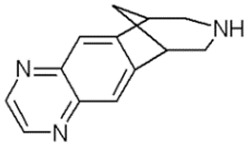	[44,56]
SCA 6	4-aminopyridine (4-AP) to ameliorate motor coordination deficiency of mouse model.	4-AP: 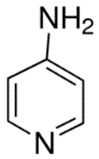	[53]
SCA 7	Interferon beta to clear mutant ataxin-7 and improve Purkinje cell survival in SCA7 knock-in mice.	Interferon beta: C_74_H_115_N_19_O_25_	[45,57]
SCA 17	Piperine (alkaloid) alleviates toxicity caused by mutant TBP protein in mouse model. NC009-1 (C_19_H_16_N_2_O_3_) reduces polyQ aggregation in Purkinje cells and ameliorated behavioral deficits in mouse model.	Piperine: 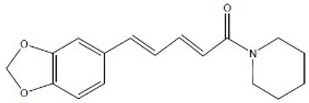 NC009-1: 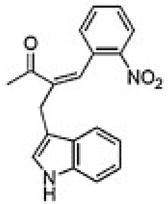	[58,59]
DRPLA	Perampanel stopped myoclonic seizures and helped to recover intellectual abilities in a 13-year-old male patient.	Perampanel: 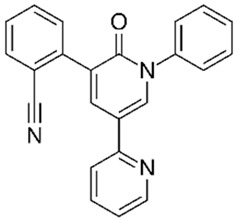	[60]

**Table 3 pharmaceutics-13-01018-t003:** Gene therapeutic strategies.

Technology	Characteristics	Functions	Advantages	Limitations	Ref.
ASOs	-Short single or double strands of chemically modified oligonucleotides. -Selectively bind to complementary mRNA.	-May induce RNase H-mediated cleavage of the targeted mRNA. -May block translation of the corresponding protein.	-Useful to knock down a gene or protein expression from RNA levels. -ASOs have favorable properties, including good distribution throughout the brain after intracerebroventricular (ICV) injection, excellent uptake by neurons and other brain cells, and high stability with a half-life of several months.	-Do not discriminate between the wild-type and the mutant alleles. -Require continuous re-administration of ASOs to offer long-term alleviation. -Lack of selectivity entails a risk of having off-target effects.	[86]
Allele-specific ASO	-Can selectively target the CAG repeat expansion.	-Specifically knocks down the mutant allele. -Required the CAG tract expansion to be associated with single-nucleotide polymorphism (SNP), to target and lower mutant allele levels.	-Maintain the wild-type protein function. -ASOs delivered into cerebrospinal fluid distribute widely throughout the central nervous system of mammals.	-Requires continuous re-administration. -CAG repeats are ubiquitous in the human transcriptome, therefore challenging. -Not all patients have the same SNP, so it is limited to a reduced number of patients.	[87]
Exon skipping (by ASO)	-ASO-based strategy aimed to remove the expanded CAG tract through alternative splicing.	-ASOs can induce exon skipping by sterically blocking the binding of splicing factors to pre-mRNAs, maintaining the RNA reading frame and rendering a truncated but functional protein.	-Global protein levels are maintained.	-Internally truncated protein is obtained. -Previous knowledge about protein translation is needed. -Exon skipping might provide a low level of protein modification.	[88]
Interferent gene silencing					
Non-allele-specific interferent gene silencing (RNA interference—RNAi)	-Cellular mechanism that induces post-transcriptional gene silencing by promoting the cleavage of target RNAs. -Implicates small RNAs (21–23 nucleotides long) that can regulate gene expression in eukaryotic organisms.	-RNAi is an evolutionarily conserved process that induces post-transcriptional gene silencing, initiated by double-stranded RNA (dsRNA) sequences, whether small interfering RNAs (siRNAs) or derived from the expression of short hairpin RNAs (shRNAs).	Potential therapeutic tool aimed to reduce or silence pathogenic gene targets, including gain of function in CNS diseases.	-Can have low effectiveness of engineered constructs at the chromosomal target, time-consuming processing, and possible undesirable mutagenic effects.	[89]
Allele-specific small interfering RNA (siRNA)	-Degradation of complementary mRNA while selectively discriminating between wt and mutated alleles.	-Uses SNPs to discriminate between WT and mutant transcripts. -This is a promissory strategy against polyQ SCA disorders.	-May use RNA duplexes that contain mismatched bases respective to the CAG target.	-Off-target effects may occur. -Poor intracellular uptake and stability in plasma. - Allele-specific gene silencing is limited to the identification of gene-linked SNPs.	[90]
Genome editing nucleases					
CRISPR-Cas9 RNA-guided nucleases (CRISPR-Cas9)	-Are used to induce targeted double-strand breaks (DSBs) at the desired chromosomal locus. -Then, non-homologous end joining (NHEJ) or homology-directed repair (HDR) is used to repair the DSB.	-Technology that uses a guide strand and a protein (Cas9) to selectively bind to a DNA region and cut. Then, both ends can bind and inactivate the gene or introduce DNA templates to edit a gene.	-Can be used to remove duplicated exons, for precise correction of causative mutation and can induce the expression of compensatory proteins. -May bring long-term efficacy. -There is no need for repeated treatment. -The expression of the modified protein is under the control of a natural promoter.	-Need more in vivo studies monitoring the off-target effects. -More studies about the potential immune responses activated by viral delivery vectors.	[91]
Transcription activator-like effector nucleases (TALENs)		-Are simple modular codes for DNA recognition. -Can act as a versatile platform for programmable DNA-binding proteins. -A FokI nuclease domain is found in TALENs.	-TALENs are simpler to construct than ZFNs. -Any DNA sequence can be targeted by TALENs, including small DNA sequences.	-Single site targeting, the occurrence of nonspecific mutations and low efficiency.	[92]
Zinc-finger nucleases (ZFNs)		-Each ZF is composed of approximately 30 aa in a conserved ββα configuration. -Then, each ZF is combined with DNA into the main channel of the DNA double helix and by a recognition of 3 to 4 bp sequence. -ZFNs are composed of 2 domains: the DNA-binding ZF protein (ZFP) domain and the FokI restriction enzyme site.	-Repair the gene sequence without the integration of any sequence into the genome. -Very high efficiency.	-Single site targeting, occurrence of nonspecific mutations and low efficiency. -Might have high immunogenic power.	[92]

**Table 4 pharmaceutics-13-01018-t004:** Different ligands used in nanoparticle coatings for brain targeting and drug delivery.

Type of Transport	Specific Target	Ligand	Examples of Nanosystems Using Ligands for Brain Delivery	Biological Effect	Ref.
Transporter-mediated transcytosis	Glucose receptors	Glucose Mannose	Paclitaxel-loaded PEG-co-poly(trimethylene carbonate) NPs modified with 2-deoxy-D-glucose	The glycosylated NPs were higher internalized compared to the NPs control. Modified NPs had high specificity and efficiency in intracranial tumor accumulation.	[177]
			Silica NPs modified with glucose and glucose-PEG- methyl ether amine	Both NP systems exhibited a significant uptake in the brain region compared with the control NPs at 1 h post-administration.	[178]
	Neutral amino acid transporter	Tyrosine Histidine Asparagine Phenylalanine threonine	Dendrimer of poly (propylene imine) coated with maltose-histidine	Maltose-histidine presence remarkably improved the biocompatibility and the ability to cross the blood–brain barrier in vivo in male wild-type mice.	[179]
	Cationic amino acid transporter	Arginine Lysine	Flurbiprofen-loaded poly (epsilon-lysine) dendrons	The penetration of the drug in bEnd.3 monolayer culture increased with the nanoformulation.	[180]
	Monocarboxylate transporter	Lactate Biotin Salicylic acid Valproic acid	Avidin-functionalized PEG- polypeptide [poly(α,β-aspartic acid) nanomicelles	Biotin targets were generated on EC surfaces. This selectively labeling promoted the targeting of avidin nanomicelles specifically to the brain microvasculature with minimal targeting into peripheral organs.	[181]
	Choline transporter	Choline Thiamine	Doxorubicin-loaded polymeric micelles modified with choline derivate	Nanocarriers treated with 20% of choline presented an enhancement in cellular uptake and antitumor activity.	[182]
Adsorptive-mediated transcytosis	Cell-penetrating peptides	Penetrating/Albumin	Triethylenetetramine-loaded liposomes functionalized with albumin or penetratin	In vivo analysis showed that surface modification remarkably increased the drug uptake into the brain tissue compared with free drug or non-modification liposome behavior.	[183]
		K16ApoE	PLGA/chitosan NPs conjugated with IgG4.1 or 125I-IgG4.1 and modified with K16ApoE by physical absorption	K16ApoE-targeted NPs were injected via femoral vein in DutchAβ_40_-treated WT mice. The results showed the accumulation of the NPs in various brain regions compared to non-modified NPs.	[184]
Receptor-mediated transcytosis	Transferrin receptor	Lactoferrin Transferrin	Clofazimine-loaded PLGA-PEG NPs modified with transferrin receptor-binding peptide	NPs presented an adequate cell interaction and high permeability across hCMEC/D3 cell monolayers.	[185]
			Dopamine-loaded mPEG-PLGA NPs modified with lactoferrin	Cellular uptake of SH-SY5Y cells and 16HBE cells improved due to lactoferrin modification of NPs.	[186]
	Endothelial LDL receptor	LDL ApoE	Doxorubicin-loaded silk fibroin/Tween 80 NPs	Tween-80 modification improved circulating time and facilitated their uptake by low-density lipoprotein.	[187]
			Rosmarinic acid-loaded polyacrylamide-chitosan-PLGA NPs functionalized with ApoE	A decrement in electrical resistance and increment in the ability to cross the BBB were observed with the concentration of ApoE increase.	[188]
	Glutathione receptor	Glutathione	Liposomal formulations (hydrogenated soy phosphatidylcholine or egg yolk phosphatidylcholine) conjugated with glutathione for methotrexate delivery	Hydrogenated soy phosphatidylcholine-glutathione liposomal increased 4-fold the drug brain delivery.	[189]

PEG: polyethylene glycol, EC: epithelial cells, PLGA: poly(lactic-co-glycolic acid), hCMEC/D3: human cortical microvessel endothelial cells/D3.

## Data Availability

Not applicable.

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
