# Peer review of "New Perspectives of Gene Therapy on Polyglutamine Spinocerebellar Ataxias: From Molecular Targets to Novel Nanovectors"

_pharmaceutics, 2021, doi:10.3390/pharmaceutics13071018_

Round 1

Reviewer 1 Report

The review manuscript, entitled “New perspectives on PolyQ-Spinocerebellar ataxias treatment: 2 From gene therapy to development of novel nanovectors,” provided a considerable summary in deciphering the clinical, pathological, physiological, and molecular aspects of the polyglutamine spinocerebellar ataxias (SCAs) and discussed potential applications of nanotechnology-based gene therapy to treat the disease. While reviews on both SCAs and nanoparticles-based brain drug delivery are not new, it is interesting to provide an overview to highlight new perspectives on nanoparticle-driven gene therapy for the treatment of polyglutamine spinocerebellar ataxias.

  1. Most introduction to nanocarriers is not related to delivering gene therapeutic agents to treat polyglutamine spinocerebellar ataxias. Provide more current studies associating with nanoparticles-based gene therapy for the treatment of polyglutamine spinocerebellar ataxias or related brain diseases.
  2. Provide a table to summarize all potent gene therapeutic agents such as Antisense, SiRNA, and CRISPR. Comment their advantages as well as limitations in the listed studies.
  3. Expand table 2 and add current studies using the transporters for brain-targeting delivery.
  4. Reorganize section head titles or subtitles: Combine “3. Current therapeutic strategies in polyQ SCAs” and “4. Therapeutic perspectives for SCAs” and rename them as “3. Pharmacological therapy treatment for polyQ-SCAs.”
  1. Highlight gene therapy by deleting “5. Gene therapy treatment for polyQ-SCAs” and changing “5.1 Antisense Oligonucleotides silencing” and “5.6 Gene edition” etc. as well as adding “interferent gene silencing” to head titles. Adding related information logically into each section.
  2. Revise the title, e.g. “New perspectives of gene therapy on polyglutamine spinocerebellar ataxias: from molecular targets to novel nanovectors.”

Author Response

The review manuscript, entitled “New perspectives on PolyQ-Spinocerebellar ataxias treatment: 2 From gene therapy to development of novel nanovectors,” provided a considerable summary in deciphering the clinical, pathological, physiological, and molecular aspects of the polyglutamine spinocerebellar ataxias (SCAs) and discussed potential applications of nanotechnology-based gene therapy to treat the disease. While reviews on both SCAs and nanoparticles-based brain drug delivery are not new, it is interesting to provide an overview to highlight new perspectives on nanoparticle-driven gene therapy for the treatment of polyglutamine spinocerebellar ataxias.

1. Most introduction to nanocarriers is not related to delivering gene therapeutic agents to treat polyglutamine spinocerebellar ataxias. Provide more current studies associating with nanoparticles-based gene therapy for the treatment of polyglutamine spinocerebellar ataxias or related brain diseases.

We thank and agree with this pertinent observation. To accomplish that, we have added pertinent references describing studies with nanoparticles for the treatment of polyglutamine spinocerebellar ataxias and related brain diseases, which would delineate current and future perspectives on gene therapy. Please see sections 6.1 to 6.4 (former 7.1 to 7.4 sections). Furthermore, we have included an additional example in section 9 (former section 10).

2. Provide a table to summarize all potent gene therapeutic agents such as Antisense, siRNA, and CRISPR. Comment their advantages as well as limitations in the listed studies.

Following this opportune suggestion, a table with the useful required information is now included in the revised Manuscript, please see Table 3, “Gene therapeutic strategies”.

3. Expand table 2 and add current studies using the transporters for brain-targeting delivery.

According to the opportune reviewer's advice, we have included additional information about the transporters for brain-targeting delivery in Table 2, with their corresponding references. Please see Table 4 (former Table 2).

5. Reorganize section head titles or subtitles: Combine “3. Current therapeutic strategies in polyQ SCAs” and “4. Therapeutic perspectives for SCAs” and rename them as “3. Pharmacological therapy treatment for polyQ-SCAs.”

As suggested, sections 3 and 4 were merged and renamed as “Pharmacological therapy treatment for polyQ-SCAs” in the revised Manuscript.

5. Highlight gene therapy by deleting “5. Gene therapy treatment for polyQ-SCAs” and changing “5.1 Antisense Oligonucleotides silencing” and “5.6 Gene edition” etc. as well as adding “interferent gene silencing” to head titles. Adding related information logically into each section.

We have made the appropriate suggested changes in head titles, as suggested by the reviewer.

6. Revise the title, e.g. “New perspectives of gene therapy on polyglutamine spinocerebellar ataxias: from molecular targets to novel nanovectors.”

In agreement with the pertinent suggestion of the reviewer, we have changed the title of the Manuscript to “New perspectives of gene therapy on polyglutamine spinocerebellar ataxias: from molecular targets to novel nanovectors”.

Reviewer 2 Report

This review provides recent experimental advances about the development of gene therapy strategies against polyQ SCAs and nanotechnology developments to treat these diseases. In the last part of the review, the application of nanoparticles-based strategies in clinical trials for the treatment of polyQ-SCA diseases are reported.

This review is well written, and all topics were adequately described. The references are updated.

To better clarify the pharmacological treatment currently available or the therapeutic perspectives for SCA, it would be better than the authors resume into two tables: 1) the current pharmacological treatments for polyQ SCAs (introducing the molecular structures of 4-aminopyridine, LD, that they mentioned and corresponding references) and 2) therapeutic perspectives for SCA.

Author Response

This review provides recent experimental advances about the development of gene therapy strategies against polyQ SCAs and nanotechnology developments to treat these diseases. In the last part of the review, the application of nanoparticles-based strategies in clinical trials for the treatment of polyQ-SCA diseases are reported.

This review is well written, and all topics were adequately described. The references are updated.

  1. To better clarify the pharmacological treatment currently available or the therapeutic perspectives for SCA, it would be better than the authors resume into two tables: 1) the current pharmacological treatments for polyQ SCAs (introducing the molecular structures of 4-aminopyridine, LD, that they mentioned and corresponding references) and 2) therapeutic perspectives for SCA.

According to the pertinent reviewer`s observation, we now present one table and one figure with the required information (molecular structures of 4-aminopyridine, LD and their corresponding references) as follow: Table 2. Current pharmacological treatments for polyQ SCAs, and Figure 2. Pharmacological therapeutic perspectives for SCA.

Round 2

Reviewer 1 Report

Accept as presented.